# DAD-Net: Classification of Alzheimer’s Disease Using ADASYN Oversampling Technique and Optimized Neural Network

**DOI:** 10.3390/molecules27207085

**Published:** 2022-10-20

**Authors:** Gulnaz Ahmed, Meng Joo Er, Mian Muhammad Sadiq Fareed, Shahid Zikria, Saqib Mahmood, Jiao He, Muhammad Asad, Syeda Fizzah Jilani, Muhammad Aslam

**Affiliations:** 1Institute of Artificial Intelligence and Marine Robotics, College of Marine Electrical, Dalian Maritime University, Dalian 116000, China or; 2Department of Software Engineering, ILMA University, Karachi 75000, Pakistan; 3Department of Computer Science, ILMA University, Karachi 75000, Pakistan; 4Department of Computer Science, Information Technology University, Lahore 54000, Pakistan; 5Department of Computer Science, Khwaja Fareed University of Engineering and Information Technology, Rahim Yar Khan 64200, Pakistan; 6School of International Business and Management, Sichuan International Studies University, Chongqing 400031, China; 7Graduate School of Information Science and Technology, University of Tokyo, Tokyo 113-8656, Japan; 8Department of Physics, Physical Sciences Building, Aberystwyth University, Aberystwyth SY23 3FL, UK; 9School of Computing Engineering and Physical Sciences, University of West of Scotland, Glasgow G72 0LH, UK

**Keywords:** Deep Learning, image classification, supervised learning, transfer learning, imbalanced data-set, mri data-set, computer-aided diagnosis, ADASYN, class activation

## Abstract

Alzheimer’s Disease (AD) is a neurological brain disorder that causes dementia and neurological dysfunction, affecting memory, behavior, and cognition. Deep Learning (DL), a kind of Artificial Intelligence (AI), has paved the way for new AD detection and automation methods. The DL model’s prediction accuracy depends on the dataset’s size. The DL models lose their accuracy when the dataset has an imbalanced class problem. This study aims to use the deep Convolutional Neural Network (CNN) to develop a reliable and efficient method for identifying Alzheimer’s disease using MRI. In this study, we offer a new CNN architecture for diagnosing Alzheimer’s disease with a modest number of parameters, making it perfect for training a smaller dataset. This proposed model correctly separates the early stages of Alzheimer’s disease and displays class activation patterns on the brain as a heat map. The proposed Detection of Alzheimer’s Disease Network (DAD-Net) is developed from scratch to correctly classify the phases of Alzheimer’s disease while reducing parameters and computation costs. The Kaggle MRI image dataset has a severe problem with class imbalance. Therefore, we used a synthetic oversampling technique to distribute the image throughout the classes and avoid the problem. Precision, recall, F1-score, Area Under the Curve (AUC), and loss are all used to compare the proposed DAD-Net against DEMENET and CNN Model. For accuracy, AUC, F1-score, precision, and recall, the DAD-Net achieved the following values for evaluation metrics: 99.22%, 99.91%, 99.19%, 99.30%, and 99.14%, respectively. The presented DAD-Net outperforms other state-of-the-art models in all evaluation metrics, according to the simulation results.

## 1. Introduction

Alzheimer’s Disease (AD) is the most frequent kind of dementia that needs substantial medical attention. Early and precise analysis of AD prognosis is required for the start of therapeutic progress, and efficient patient therapy [1]. According to a study, 10 million new cases of dementia are registered every year [2]. The World Health Organization (WHO) reported that AD had surpassed cancer as the fifth most significant cause of death, with the number of AD patients expected to reach 152 million by 2050 [2]. AD is a long-term neurological brain disease that gradually destroys brain cells, causing memory loss and cognitive problems and finally accelerating the loss of ability to perform day-to-day activities of real-life [3].

AD is a brain-neurological degeneration disorder [4]. It is categorized as dementia, atrophy of the human brain affecting memory, and causes loss of behavioral, social, and reasoning faculties. It is caused by the accumulation of protein fragments in the brain [5,6,7]. Plaques and tangles are formed around the neurons inside the human brain, resulting in abnormal lobes, hippocampus shrinking, and enlarged ventricles [8]. It is an incurable fatal disease [9,10] with a lifetime of agony for the patient and a severe mental, physical, and financial toll of suffering for the patient’s family. The cause of AD is unknown, and there are no effective medications or therapies to reverse dementia. Mild Cognitive Impairment (MCI), a pre-clinical stage of AD, is a transitory state between normal ageing and AD.

Detecting the risk and severity of AD at its early stages is very critical [11,12]. However, doctors can classify AD in its early stages using neuro-imaging and computer-assisted diagnostic approaches with significantly less accuracy. Neuro-imaging, including Computed Tomography (CT) scan, Positron Emission Tomography (PET) scan, and specifically Magnetic Resonance Imaging (MRI) scan, play a vital role in medical diagnosis [4]. It is an effective non-invasive method that provides information about the human body. The advancement of the medical diagnosis process has created tremendous research trends in computer-aided diagnosis nowadays [5,6].

Over the years, numerous Machine Learning (ML) [13] and Deep Learning (DL) [14] algorithms have been developed by many researchers around the globe for AD detection and classification. Many researchers have achieved remarkable results using the DL algorithms; however, there is still room for improvement. In this series of DL models, a hybrid Convolutional Neural Network (CNN), a CNN model with slice selection, and a CNN model with histogram stretching are introduced in [15,16,17]. Others proposed a CNN model with skull striping [18] and a CNN model which utilized the slicing samples for pre-processing is introduced in [19]. However, the focus of these deep models is primarily biased towards classification due to the black-box nature of CNN.

In the literature on AD, some researchers have developed miscellaneous tools and applications for automated segmentation of neuro-images [20]. These applications are Vol-Brain [21], and Fusion of neuro-imaging Pre-processing [22]. Although these applications are practical tools for segmenting neuro-image, the research focused on visualizing the classification process through CNN layers is scarce. The feature map of each convolution layer reveals various filters being applied to the image, and it provides a hint to what sort of filters the model uses on the image for feature extraction [23]. This approach supports grad-CAM [24] heat-map, which shows class activation via gradient-based localization map. The proposed CNN model uses a series of conventional blocks consisting of different deep layers to accomplish outstanding classification results. The goal of the proposed DAD-Net is to obtain an accurate classification result for detecting AD in its earlier stages with better accuracy. The main contributions of the research study are:We propose a new convolutional neural network architecture for detecting AD with relatively few parameters, and the proposed solution is ideal for training a smaller dataset.The previous methods [22,23,25] accuracy compromised on Alzheimer’s data-set due to an imbalanced number of classes. To handle the imbalance problem of the Alzheimer’s data-set, we exploited the ADASYN oversampling algorithm, which interpolates new images to balance the class samples.The Grad-CAM heat-map algorithm is utilized to visualize the class activation map, highlighting the features that lead to the classification of an image sample.The proposed model is extensively compared with several other approaches using various evaluation parameters: Accuracy, AUC, Precision, Recall, F1-score, and size of trainable parameters. It is observed that our approach outperforms other state-of-the-art models.

The rest of the paper is arranged in the following way: The related studies are briefed in Section 2. The methodology and proposed DAD-Net model for AD classification details are presented with the description of the dataset and model components in Section 3. The visualization process and the DAD-Net model evaluation with the state-of-the-art model are presented in Section 4 and the last section, Section 5 concludes the paper with future goals.

## 2. Related Work

Precise classification of medical images is a strenuous task because of the complicated procedure of obtaining medical data-sets [25]. Unlike other data-sets, medical data-sets are prepared by expert specialists and contain sensitive and private information of patients which cannot be publicly disclosed to anyone. That is why organizations and institutions such as Alzheimer’s Disease Neuroimaging Initiative (ADNI) [26] and Open Access Series of Imaging Studies (OASIS) [27] providing medical data-sets have a screening process for accessing their data-sets which requires an application to be filled and terms to be agreed by the researcher, constraining them to using it for research purposes only [28,29,30,31]. Medical data-sets are inherently highly imbalanced because it is impossible to compile a data-set with an equal number of patients with health and ailment samples. The techniques to tackle this problem are pretty challenging themselves [32,33,34]. OASIS data-set containing 416 3D samples is used by Jyoti et al. [35] to create a CNN model with the convolution layer, batch normalization layer, pooling layer, and Adam optimizer.

To evaluate their model accuracy, the authors compared their model with two different pre-trained architectures such as InceptionV4 [36] and ResNet [35]. A cost-sensitive training technique is used to overcome the data-set imbalance problem is discussed in [37]. The cost matrix modified the result of the output layer to give more importance to classes with fewer samples, and the experiments achieved a precision of 75%. A comparative analysis of state-of-the-art Alzheimer’s disease classification models is depicted in Table 1. The traditional deep and transfer learning models with novel approaches achieve outstanding accuracy greater than 90%.

A similar approach is adopted by Hussain et al. [37] for the same OASIS data-set. They used a 12-layer CNN architecture, including convolution and pooling operations. They used Leaky ReLU [37] in combination with MaxPooling as activation function instead of ReLU [39] to avoid gradient vanishing issue [40]. The authors compared their model with four different pre-trained models such as InceptionV3, Xception [41], MobileNetv2 [42], and VGG19 [25] to analyze the performance of the model. The model achieves an accuracy of 97.75% during experiments compared to pre-trained models.

The same data-set from Kaggle is used by Sugnathe et al. [16] to implement a hybrid framework using ResNet V2 with Inception V4. In this model, the ResNet V2 integrates residual connections to the pre-trained Inception V4 model [37]. In the experiments, the model is assessed by varying learning rates and optimizers, producing the highest accuracy of 79.12%. Pradhan et al. [31] perform a simple comparative study using two state-of-the-art pre-trained models such as VGG19 and DenseNet169 [43]. These two models are selected due to the ability of VGG19 to train on many classes with remarkable accuracy, and the DenseNet169 can handle vanishing gradient issues and reduce the number of training parameters. The data-set from Kaggle was fed to both models via Image Data Generator (IDG) with different augmentation parameters. Through the augmentation, the pre-trained models such as VGG19 and DenseNet169 achieved an accuracy of 88% and 87%, respectively. Bettini et al. [33] employed an OASIS-3 data-set and created a five-layer CNN model to classify three different early stages of Alzheimer’s disease [44].

Not all the features extracted by a deep model are helpful in accurately predicting the correct class of a sample. Some features hinder a model from reaching desired results [45,46]. This issue of deep models was tackled by El-Aal et al. [29] and presented a novel approach to selecting specific features from the feature map of deep models, which ultimately improves the classification results and reduces the training time of the model. The ResNet101 and DenseNet201 for feature extraction, while the Rival Genetic Algorithm (RGA) [47] and Probability Binary Particle Swarm Optimization (PBPSO) [48] algorithms were used for feature selection. The selected and control features were fed to a separately created classification model. ResNet101 and DenseNet201 provided the best results with PBPSO and achieved an accuracy of 87.3% and 94.8 %, respectively.

Raju et al. [38] utilized a class activation heat-map algorithm named Grad-CAM, which uses gradient data for its calculations, and heat-maps to help understand the working of a deep model. They selected a transfer learning approach for training a deep model and modified the VGG16 by adding an extra dense layer at the end of the model. The model’s performance is enhanced by Fastai [49] using the grad-CAM to highlight the regions of the brain on MRI samples that the previous model selects for making predictions. Dozat et al. [50] suggested that SGD loss function in combination with Nesterov intensity to further improved the classification results, and the model attained a test accuracy of 97.89%.

There is always scope for improvement in DL, and most researchers have not achieved remarkable classification performance. Their methodologies and approaches suffer from various hindering factors because they have overlooked some inherent hurdles of DL models and medical image data-sets [5,12,23]. The data-set used in this research is collected from Kaggle, which contains 6400 samples of anonymous patients with only MRI scan images and their respective class labels information. It is a multi-class data-set consisting of four different classes, including a customary (NOD) class and three other classes representing three different early stages of AD, namely, Very Mild Demented (VMD), Mild Demented (MD), and Moderate Demented (MOD). It is a two-year-old data-set, and various researchers have offered their contributions in this duration while obtaining good results by employing several techniques and combinations.

## 3. The Proposed Model for Early Alzheimer Diagnosis

In the medicine and healthcare field, image processing has brought quite a revolution. Nowadays, image processing has applications in almost every aspect of the medical field. Doctors can examine the organs of the human body from the inside without needing surgery during the diagnosis stage. There are various types of scans in the medical field: X-ray, Ultrasound, Magnetic Resonance Imaging (MRI), and Computer Tomography (CT) scans. A human being cannot possibly examine medical scans as precisely as a machine is capable of and draw accurate conclusions from them. A device trained on a medical image data-set can provide accurate results within seconds, whereas, on the other hand, it might take a whole panel of doctors to derive the same conclusion in days. Modern health care systems depend upon computer vision and image processing algorithms as their integral part. The importance cannot be overstated. AD is becoming one of the most rapidly increasing diseases globally. A few researchers have used data augmentation techniques to improve their results [51,52]. In contrast, none of the reviewed research papers regarding the classification of Alzheimer’s disease has recognized the central problem of data-set imbalance. Some researchers failed to obtain notable results because they did not train their models enough. It is observed that research papers focus on discovering new approaches toward classification purposes for biomedical diagnoses.

In this proposed model, the input data-set is pre-processed using normalization and the essential process of converting the categorical data variables is to be provided to the DAD-Net using the one-hot encoder. Then, the Adaptive Synthetic (ADASYN) algorithm is utilized to solve the imbalanced data-set issue that over-samples the classes to balance the data-set. Afterward, the data-set is split into train, test, and validation by 80%, 10%, and 10%, respectively. Furthermore, the features are extracted using a standard CNN for effectively training the DAD-Net as shown in Figure 1. The size of training parameters is smaller in comparison with [29,31,33] for the robustness of the model in AD classification. The Grad-CAM heat-map algorithm is utilized to visualize the class activation map, highlighting the features that lead to the classification of an image sample.

### 3.1. Description of the AD Dataset

Several data-sets are available on the internet for AD classification. Many AD data-sets are in CSV format and are unsuitable for this research. Dedicated organizations such as ADNI and OASIS also provide access to their data-sets for research and educational purposes. However, the samples in both of these data-sets are in three-dimensional image format, and the size of the data-sets is gigantic. The data-set is 18 gigabytes, while the ADNI dataset is 450 gigabytes. The data-set used in this research is collected from Kaggle, which contains samples of anonymous patients with only MRI scan images and their respective class labels information. It is a multi-class data-set consisting of different views and four classes, including an average NOD class and three other classes representing three different early stages of AD. VMD, MD, and MOD are slightly observable with a bare eye in Figure 2.

According to the description of the dataset, each sample in the data-set available on Kaggle is personally verified by the uploader himself. Furthermore, the data-set size is reasonable, and the pieces are already cleaned up, i.e., resized and organized. Based on these factors, this data-set is used in our research. The data-set has 6400 samples in total. The samples are individual three-channel (RGB) images of 150 × 150 pixels belonging to four different classes. The number of samples in the NOD class is 3200. The remaining three classes, VMD, MD, and MOD, have 2240, 896, and 64 images, respectively. The only downside of this data-set is that it is imbalanced, as discussed in Figure 3. To solve this problem, we use ADASYN to generate synthetic data for each imbalance class concerning the balanced class, as shown in Figure 4. The data-set is divided into 80%, 10%, and 10% for training, validation, and test set, respectively.

#### 3.1.1. Balancing the AD Dataset using ADASYN

Oversampling and undersampling are two standard re-sampling techniques. The minority class samples are generated using oversampling, whereas the majority class samples are reduced using undersampling. In the suggested method, however, we employ an oversampling method ADASYN [53] to balance all the classes and provide more accurate and exact AD classification findings. The primary idea behind ADASYN is to use a weighted distribution for different minority class examples based on their learning difficulty, with more synthetic data created for minority class examples that are harder to learn than minority class examples that are easier to learn. As a result, the ADASYN technique improves data distribution learning in two ways: first, it reduces bias caused by class imbalance, and second, it adaptively pushes the classification decision boundary toward challenging samples. As shown in Figure 4, ADASYN efficiently balances the samples of each class and solves the dataset imbalance problem. ADASYN uses the Nearest Neighbor approach to interpolate additional imitation examples for the minority classes exhibited to balance the data-set as depicted in Figure 5.

#### 3.1.2. The Proposed DAD-Net Network Architecture

The CNN architecture is based on the biological structure of the human brain, and it is mainly used in computer vision applications such as image classification, image segmentation, and object detection. Previously designed deep models preferred it due to its translation-invariant nature [47]. The translation or space invariance implies that a CNN can recognize the same feature regardless of its position in various images. This paper proposes a novel CNN model from scratch to perform accurate AD classification. The proposed DAD-Net is comprised of five convolutional blocks, and each convolutional block has a Rectified Linear Unit (ReLU) activation function, kernel initializer LecunUniform V2 and a 2D Max pooling layer, one dropout layer, one dense layer, and a SoftMax classification layer as depicted in Figure 6. The proposed model’s detailed network architecture and model summary for classifying AD with the subsequent layer is discussed in Table 2. The main components of the proposed model are briefly discussed in the following subsections.

#### 3.1.3. DAD-Net Convolutional Blocks

The convolutional block is the main block of the proposed DAD-Net, and each convolutional block consists of a convolutional 2D, a ReLU, and a Max-pooling2D layer. The kernel initializer LecunUniform V2 is used to choose weights for the convolutional 2D layer kernel weights. The ReLU activation function is used to overcome the gradient vanishing problem and allow the network to learn and perform faster. At the same time, the convolutional 2D down-samples the image and its spatial dimensions by taking the maximum value over an input window (of size defined by pool_size) for each channel of the input. The convolutional layers work in asymmetry, and the features are gradually built. Local patterns such as edges, lines, and curves are extracted in the initial layers, and local features are extracted based on these patterns. The model extracts low-level, mid-level, and high-level features and enables the deep model to classify an image accurately.

#### 3.1.4. Dropout Layer

Dropout layers turn nodes on and off to reduce the training time of the model and decrease the network complexity. The dropout layer randomly switches off nodes using probability distribution during each epoch, preventing models from over-fitting. As a result, the model learns all the relevant features and contains various features in each iteration.

#### 3.1.5. Flatten Layer

Flatten layer is placed between the convolution layers and dense layers. Convolution layers work with tensor data types for input, while dense layers require input in a one-dimensional format. Flatten layer vectorizes the feature map to feed it to dense layers, as depicted in Figure 7.

### 3.2. Dense Block

The proposed architecture has two dense layers and the details of the dense layers and remaining layers are discussed in the following subsection.

#### 3.2.1. ReLU Activation

Activation functions are mathematical operations that decide whether output from a perceptron is to be forwarded to the next layer. In short, they activate and deactivate nodes in a deep model. The activation function is used in the output layer to activate the node, which returns its label, which is then assigned to the image processed through the model. There are several activation functions. We used ReLU in hidden layers because of its simple and time-saving calculation. SoftMax, a probability-based activation function, is used for the output layer because our model is for multi-class classification.

#### 3.2.2. Dense Layer

The dense layer is also called the fully connected layer. This layer inputs a single vector and produces output based on its parameters. The images are identified and assigned a class label in these layers. The learning of the model takes place in fully connected layers via the back-propagation method. The number of trainable parameters of a model is determined based on the number of values used in each dense layer. SoftMax is used after a couple of layers, with the number of neurons equal to the number of classes [48]. The labels are one-hot encoding in multi-class classification, and only the positive type is present in the loss term.

## 4. Evaluation of the Proposed DAD-Net

The experiments were executed on a personal computer system equipped with two Intel Xeon 2687W v4 (3.0 GHz clock speed, 12 cores, and 24 threads) CPUs, 64 GB RAM, 5 GB (NVIDIA) P2000 GPU (Graphical Processing Unit). The model’s evaluation was conducted using the test set created from splitting the data-set before training the model. Using several metrics ensures the robustness of a model from every angle. The combined understanding of these results determines the successful training of a model. For instance, if accuracy is very high, say above 90% does not necessarily mean that the model is excellent. Several other factors are involved, such as loss, over-fitting, etc. We employed different metrics to benchmark the performance of our model. The following six terms are extensively used when observing various metrics of a classifier.

### 4.1. Accuracy

Accuracy is the measure of total correct predictions out of real predictions obtained using the following expressions:(1)$$\begin{array}{c} Accuracy=\left(\frac{TP+TN}{TP+FN+FP+TN}\right) \end{array}$$
where, *TP*, *TN*, *FN*, *FP* are True Positive, True Negative, False Negative, and False Positive values, respectively.

### 4.2. Precision

Precision is the ratio of correct positive predictions to total positive predictions and it is calculated using the following equation:(2)$$\begin{array}{c} Precision\;\;=\left(\frac{TP}{TP+FP}\right) \end{array}$$

### 4.3. Recall

The recall is also known as the sensitivity score or actual positive rate. It is the comparison of correct optimistic predictions to total real correct positives. The recall is calculated using the following equation:(3)$$\begin{array}{c} Recall\;\;=\left(\frac{TP}{TP+FN}\right) \end{array}$$

### 4.4. F1-score

Ideally, a value of 1.0 in precision and 1.0 in the recall is considered an ideal case for a classification model. *F*1-score is the harmonic mean of precision and recall. *F*1-score is unique in the sense that it plots its graph with a separate line for each class label. The *F*1-score is computed using the following equation:(4)$$\begin{array}{c} F1\;\;=\left(2\times \frac{Precision\;\times \;Recall}{Precision\;+\;Recall}\right) \end{array}$$

### 4.5. Receiver Operating Characteristics (ROC) Curve

A ROC curve is a graphical way to illustrate the possible connection between sensitivity and specificity for every possible cut-off for a combination of tests. The ROC-curve graph is illustrated with the help of 1–specificity (on the *x*-axis) and sensitivity (on the *y*-axis). While the 1–specificity is False Positive Rate and sensitivity is True Positive Rate that can be obtained through the following expressions:(5)$$\begin{array}{c} TPR=\left(\frac{FP}{FP+FN}\right) \end{array}$$
(6)$$\begin{array}{c} FPR=\left(\frac{FP}{FP+TN}\right) \end{array}$$

### 4.6. Confusion Matrix

The confusion matrix is used to assess and calculate different metrics of a classification model. It provides the division of number all the predictions a model has made during the training or testing phase.

### 4.7. DAD-Net Model Self-Comparison with Accuracy, F1-Score, Extension Receiver, and AUC

An ROC curve is used to analyze the performance of clinical tests and, more specifically, the accuracy of a classifier for binary or multi-classification. The Area Under Curve (AUC) in a ROC curve is used to measure the usefulness of the classifier, where greater the AUC generally means greater the usefulness of the classifier. We check the usefulness and accuracy of our proposed DAD-Net model using AUC and extension of ROC as depicted in Figure 8 for each class separately using AD data-set with and without ADASYN. The proposed DAD-Net model has AUC = 98.41% and AUC = 99.91% without and using ADASYN, respectively. We can note that after balancing the AD data-set using the ADASYN algorithm, the AUC significantly changed from 98.41% to 99.91%, as depicted in Figure 9 and this similar effect has also been noted in AUC for all the classes. The AUC of class 0 (MD), class 1 (MOD), class 2 (ND), and class 3 (VMD) is 86.53%, 81.25%, 92.01%, and 88.50%, respectively without balancing the data-set as depicted in After balancing the AD data-set, The AUC of class 0 (MD), class 1 (MOD), class 2 (ND), and class 3 (VMD) is 99.99%, 99.99%, 99.99%, and 98.94%, respectively. Moreover, the Accuracy and F1-Score of the proposed model with ADASYN are significantly improved, as depicted in Figure 10 and Figure 11. These improvements in Accuracy, AUC, F1-Score and Extension receiver prove the authenticity of the ADASYN algorithm and feature selection of the DAD-Net model.

### 4.8. The Proposed Model Comparison with Recent Models Using ROC

ROC curve is used to analyze the performance of clinical tests and, more specifically, the accuracy of a classifier for binary or multi-classification. The Area Under Curve (AUC) in a ROC curve is used to measure the usefulness of the classifier, where greater the AUC generally means greater the usefulness of the classifier. We check the effectiveness and accuracy of our proposed DAD-Net model using the ROC curve using the AD data-set with ADASYN. The proposed DAD-Net is compared using the ROC curve with DEMENT, CNN Model, on the AD dataset. The proposed DAD-Net, DEMNET, CNN Model, after balancing the AD dataset with ADASYN, achieved ROC values of 99.02%, 98.33%, 97.68%, respectively on the balanced AD dataset as depicted in Figure 12.

### 4.9. DAD-Net Comparison with Other Models Using Extension of ROC for Multi class

ROC curves are commonly used in binary classification to investigate a classifier’s output. Binarizing the output is required to expand the ROC curve and ROC area to multi-class or multi-label classification. One ROC curve can be generated for each label, however, each element of the label indicator matrix can also be treated as a binary prediction (micro-averaging). The proposed DAD-Net is compared using the Extension of the ROC curve with DEMNET, CNN Model, on the balance AD dataset as depicted in Figure 13. We can note that after balancing the AD data-set using the ADASYN algorithm the AUC significantly improved for all the approaches. This similar effect has also been noted in AUC for all the classes of the proposed DAD-Net. The AUC after balancing the AD data-set, the AUC of class 0 (MD), class 1 (MOD), class 2 (NOD), and class 3 (VMD) is 99.99%, 99.99%, 99.02%, and 98.94%, respectively. These improvements in AUC prove the authenticity of the ADASYN algorithm and feature selection of the DAD-Net model.

### 4.10. Accuracy Comparison against Other Model with and without ADASYN

ADASYN algorithm is applied to the data-set to up-sample the number of images in classes with fewer samples. It increased the size of the data-set from 6400 samples to 12,788 instances as discussed above. Hence, balancing out the data imbalance problem. The contrast between the two methods is utilizing the up-sampling technique, ADASYN. For a fair comparison, we evaluated our proposed and recent models such as DEMNET and CNN Model, using the same AD dataset after balancing it through ADASYN. The system provides remarkable results with ADASYN for the proposed and other models. The proposed DAD-Net model, DEMNET, and CNN Model achieved accuracies of 99.22%, 98.67%, and 98.10%, respectively, using the balanced AD data-set. This significant improvement in accuracies of all the models is visible from Figure 14.

### 4.11. Comparison of DAD-Net with Recent Models Using F1-Score

The input data-set is normalized in this suggested DAD-Net model. The fundamental procedure of converting categorical data variables is delivered to the model utilizing the one-hot encoder. The ADASYN technique is then used to correct the unbalanced data-set problem by oversampling the classes to balance the data-set. On the AD data-set, we evaluated the DAD-Net model with recent models such as DEMNET and CNN Model for a fair comparison. The system using ADASYN produces remarkable results for the suggested model and other models. Using a balanced AD data-set, the proposed DAD-Net model, DEMNET, and CNN Model F1-score of 99.19%, 98.67%, and 98.15%, respectively. This significant improvement in accuracies of all the models is visible from Figure 15.

### 4.12. Comparison of DAD-Net with Recent Models Using Precision

Several deep models were developed to classify Alzheimer’s disease in its early stages. Some algorithms were traditional CNN, while others were pre-trained deep architectures. As mentioned earlier in this paper, our proposed model is a deep CNN-based DAD-Net comprising distinct Conv-blocks. We compared our model with the DEMNET and CNN classification model. The first model is a framework of DEMNET with a precision value of 98.67% after balancing the AD data-set using ADASYN. The second model was built with CNN Model, and its evaluation precision value is 98.21% on balanced tasks, respectively, as shown in Figure 16. The proposed model achieved a precision value is 99.30% AD datasets. As a result of the preceding discussion, we discovered that the presented model’s performance is better and more consistent than models in the form of precision.

### 4.13. Comparison of DAD-Net with Recent Models Using recall

The recall is determined by dividing the total number of positive samples by the number of positive samples accurately categorized as positive. The memory measures the model’s ability to recognize positive samples. The higher recall values represent the more significant number of positive samples found. DAD-Net is a system of convolutional layers compared with the DEMNET and CNN classification models. The first model is a framework of DEMNET with a recall value of 98.59% after balancing the AD data-set using ADASYN. The second model was built with CNN Model, and its evaluation recall value is 98.13% on balanced tasks, respectively, as shown in Figure 17. The proposed model’s achieved recall value is 99.14% on AD datasets. The result of the preceding discussion is that the presented models perform remarkably in the form of recall.

### 4.14. AUC Comparison of Proposed Model with other Models

Our proposed model is a deep CNN-based DAD-Net consisting of different convolution blocks and is very effective in classifying the other AD classes, as discussed earlier in this paper. We also compared our DAD-Net with state-of-the-art classification models such as DEMENT and CNN. The DEMNET had achieved an AUC of 99.94% after balancing the AD data-set through ADASYN as depicted in Figure 18. The second model is created using CNN, and its AUC score is 99.96% on the balanced AD dataset. The proposed model attained an AUC value of 99.91% on both AD datasets, as depicted in Figure 18. As a result of the above discussion, we noted that the performance of the proposed model remains better and more consistent in comparison with hybrid models in the form of AUC.

### 4.15. Comparison of DAD-Net with Recent Models Using Confusion Matrix

In this proposed DAD-Net model, the input data-set is pre-processed using normalisation and the essential process of converting the categorical data variables to be provided to the model using the one-hot encoder. The confusion matrix of DEMNET, CNN Model, and Proposed DAD-Net are shown in Figure 19.

### 4.16. Visualization through Gradient-Weighted Class Activation Map

Grad-CAM detects the discriminatory regions for a CNN classification by calculating its CAM using gradient data. Grad-CAM visualizes a map of all the working classes by integrating gradient information. Grad-CAM considers 2D activation’s along with the average gradient information. It supports recognizing what a network perceives, and which particular neuron is firing in a specific deep layer [47]. The preceding class gradient is related to the channel, ensuring the last CNN layer to generate a localization CAM displaying the important locations in the image that has a substantial effect on the deep model’s prediction as shown in Figure 20. To generate the CAM, the class gradient score is computed relative to the feature maps of the CNN layers [47]. All the simulation results using different quality metrics are evident of the performance of deep DAD-NET. The detailed comparison of DAD-Net and other deep models is discussed in Table 3.

## 5. Conclusions

In this paper, we proposed a novel deep CNN for detecting AD with relatively few parameters, and the proposed solution is ideal for training a smaller dataset. The Detection of Alzheimer’s Disease Network (DAD-Net) is built from scratch to precisely classify the stages of AD by decreasing parameters and calculation costs. Each block is specifically designed with many layers named Conv-block, which is used to classify the AD in its early stages for all the specific classes. ADASYN method is employed for handling data-set imbalance problems for generating new instances to balance the number of samples for each category. Grad-CAM algorithm provides insight into CNN layers’ working by visualizing class activation heat-map. Our proposed deep model provides outstanding accuracy of 99.22%, 99.30% precision, sensitivity (Recall) of 99.14%, and an impressive AUC value of 99.91%. We will involve other pre-trained architectures and fine-tune transfer learning models to achieve more desirable results in the future.

## Figures and Tables

**Figure 1 molecules-27-07085-f001:**
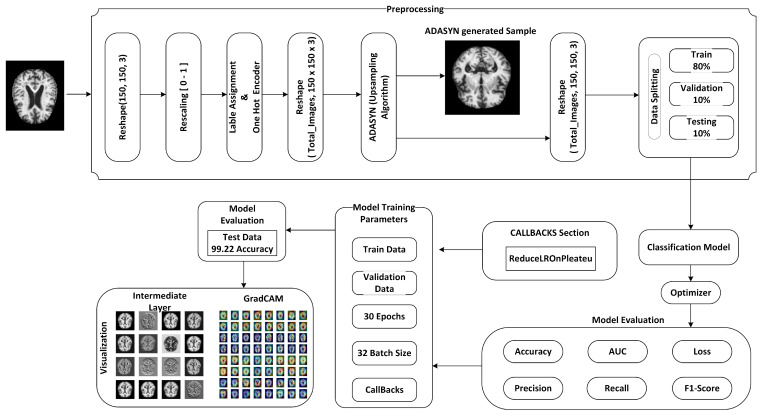
Methodology of the proposed DAD-Net that is used for early detection of AD.

**Figure 2 molecules-27-07085-f002:**
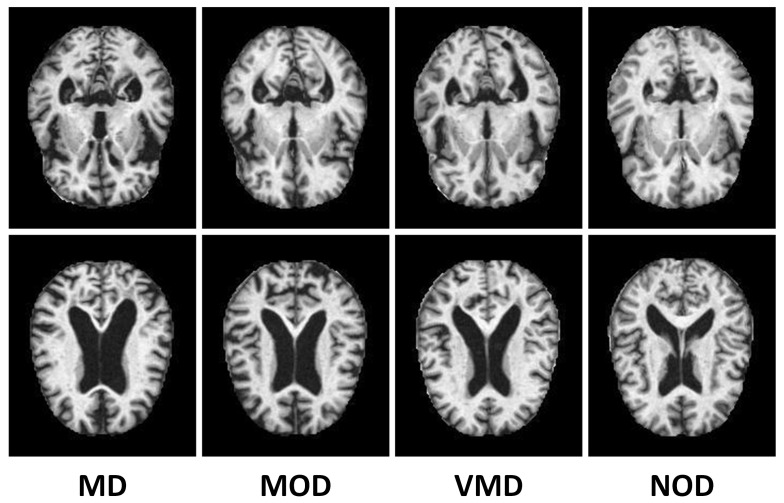
The image samples from AD dataset without up-sampling through ADASYN.

**Figure 3 molecules-27-07085-f003:**
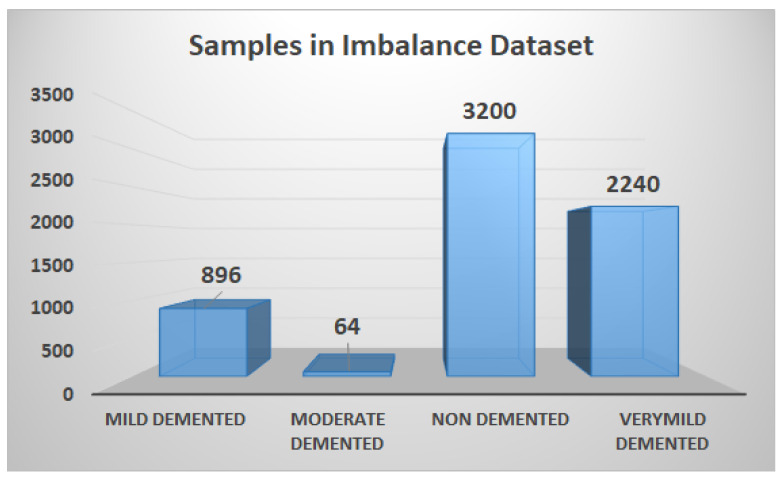
AD data-set class distribution before up-sampling through ADASYN.

**Figure 4 molecules-27-07085-f004:**
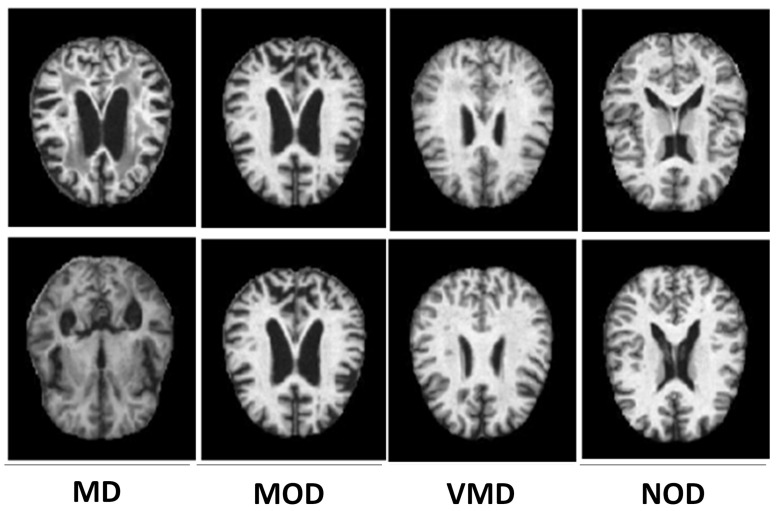
The synthetic image samples generated through ADASYN for all classes.

**Figure 5 molecules-27-07085-f005:**
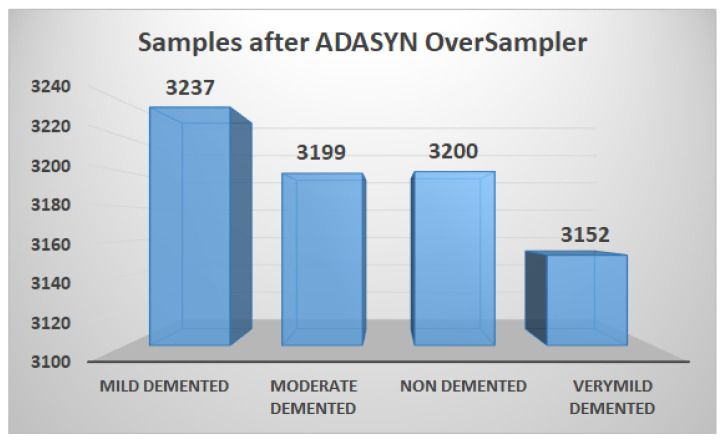
AD data-set class distribution after up-sampling through ADASYN.

**Figure 6 molecules-27-07085-f006:**
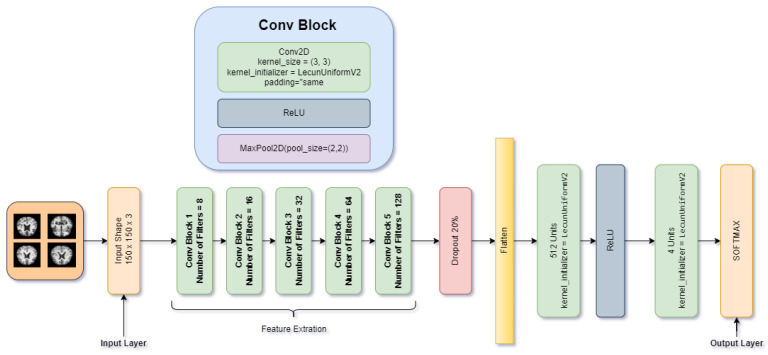
Detailed architecture of the proposed DAD-Net for early detection of AD.

**Figure 7 molecules-27-07085-f007:**
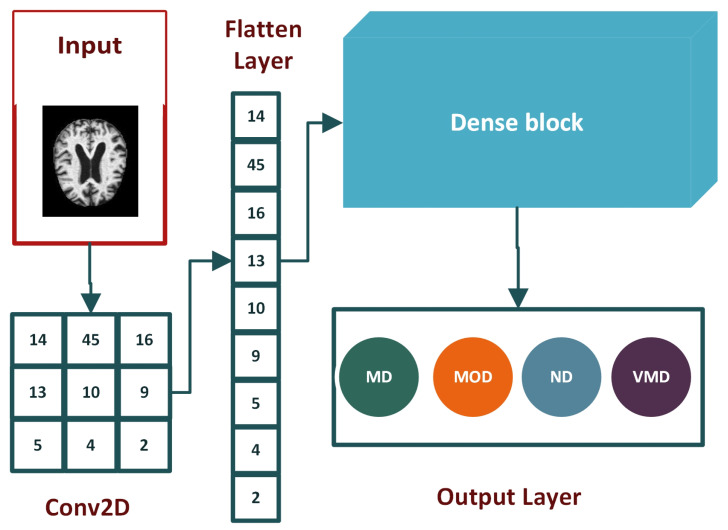
The overview of the architecture of the flattened layer used to vectorize the feature map.

**Figure 8 molecules-27-07085-f008:**
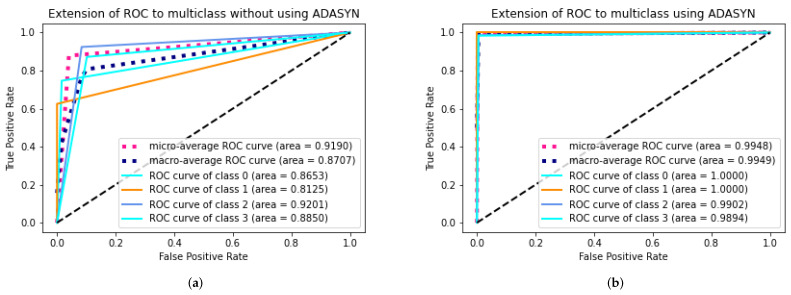
Self-comparison of Extension receiver of proposed DAD-Net With and without using ADASYN up-sampling algorithm. (**a**) Proposed DAD-Net without ADASYN; (**b**) Proposed DAD-Net with ADASYN.

**Figure 9 molecules-27-07085-f009:**
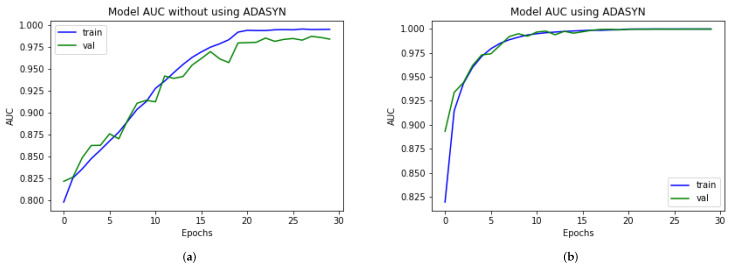
Self-comparison of AUC of proposed DAD-Net With and without using ADASYN up-sampling algorithm. (**a**) Proposed DAD-Net without ADASYN; (**b**) Proposed DAD-Net with ADASYN.

**Figure 10 molecules-27-07085-f010:**
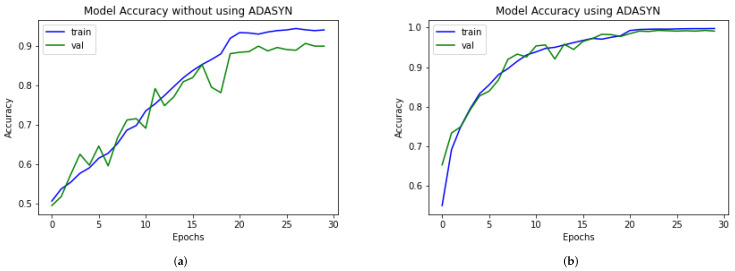
Self-comparison of the accuracy of proposed DAD-Net With and without using ADASYN up-sampling algorithm. (**a**) Proposed DAD-Net without ADASYN; (**b**) Proposed DAD-Net with ADASYN.

**Figure 11 molecules-27-07085-f011:**
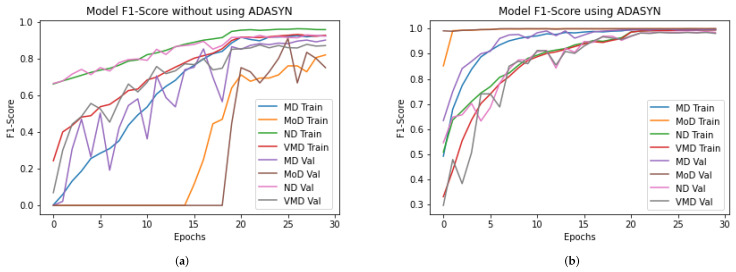
Self-comparison of F1-score of proposed DAD-Net With and without using ADASYN up-sampling algorithm. (**a**) Proposed DAD-Net without ADASYN; (**b**) Proposed DAD-Net with ADASYN.

**Figure 12 molecules-27-07085-f012:**
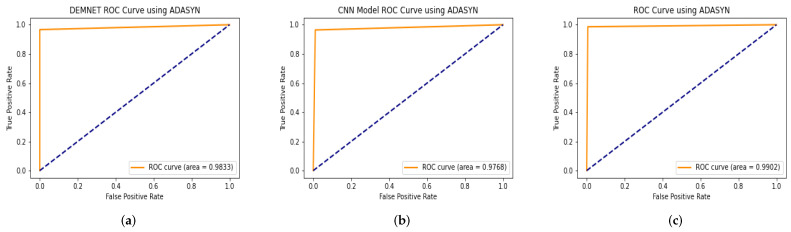
ROC Curve results of DEMNET, CNN Model, and DAD-Net using ADASYN. (**a**) DEMNET; (**b**) CNN Model; (**c**) Proposed DAD-Net.

**Figure 13 molecules-27-07085-f013:**
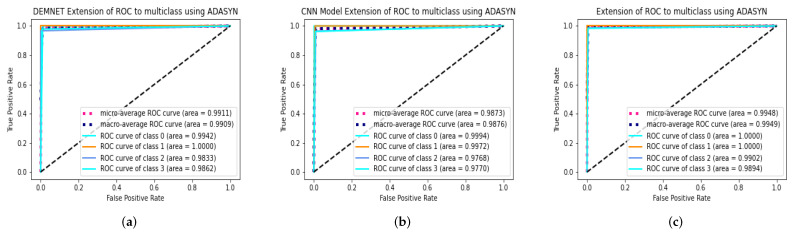
Extension Receiver results of DEMNET, CNN Model, and DAD-Net using ADASYN. (**a**) DEMNET; (**b**) CNN Model; (**c**) Proposed DAD-Net.

**Figure 14 molecules-27-07085-f014:**
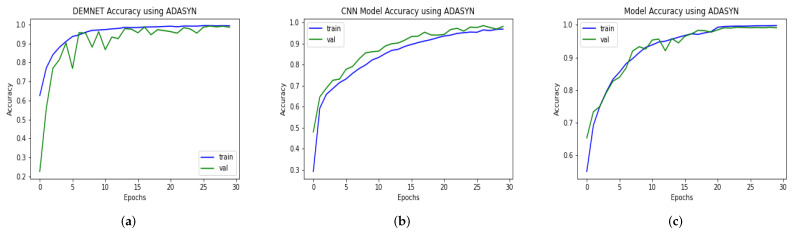
Accuracy Comparison of DEMNET, CNN Model, and DAD-Net using ADASYN. (**a**) DEMNET; (**b**) CNN Model; (**c**) Proposed DAD-Net.

**Figure 15 molecules-27-07085-f015:**
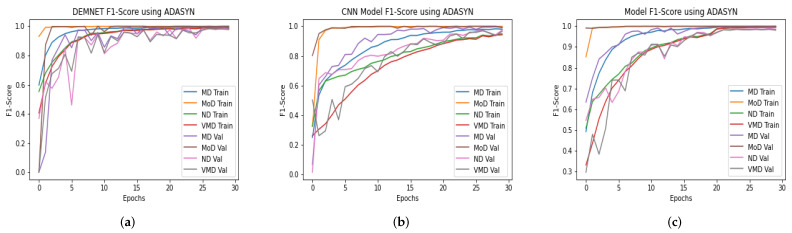
F1-Score Comparison of DEMNET, CNN Model, and DAD-Net using ADASYN. (**a**) DEMNET; (**b**) CNN Model; (**c**) Proposed DAD-Net.

**Figure 16 molecules-27-07085-f016:**
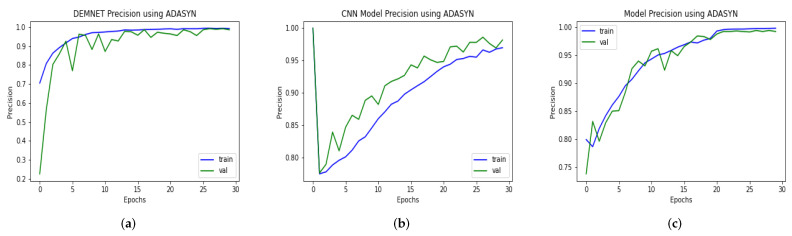
Precision Comparison of DEMNET, CNN Model, and DAD-Net using ADASYN. (**a**) DEMNET; (**b**) CNN Model; (**c**) Proposed DAD-Net.

**Figure 17 molecules-27-07085-f017:**
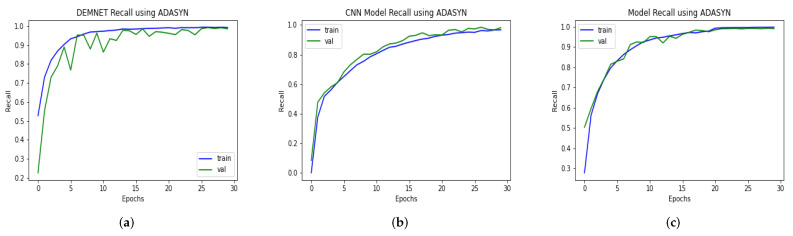
Recall Comparison of DEMNET, CNN Model, and DAD-Net using ADASYN. (**a**) DEMNET; (**b**) CNN Model; (**c**) Proposed DAD-Net.

**Figure 18 molecules-27-07085-f018:**
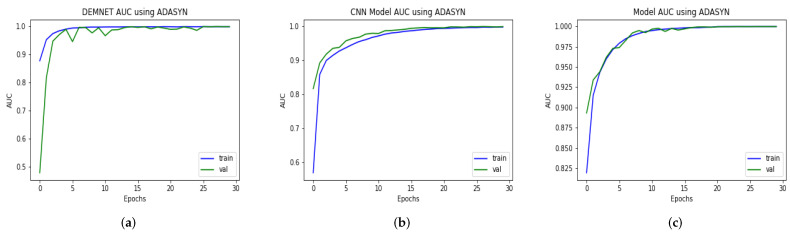
AUC Comparison of DEMNET, CNN Model, and DAD-Net using ADASYN. (**a**) DEMNET; (**b**) CNN Model; (**c**) Proposed DAD-Net.

**Figure 19 molecules-27-07085-f019:**
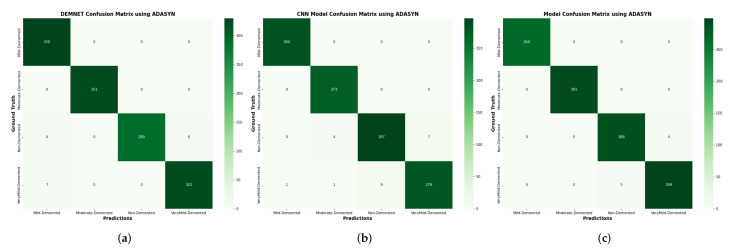
Confusion matrix of DEMNET, CNN Model, and DAD-Net using ADASYN. (**a**) DEMNET; (**b**) CNN Model; (**c**) Proposed DAD-Net.

**Figure 20 molecules-27-07085-f020:**
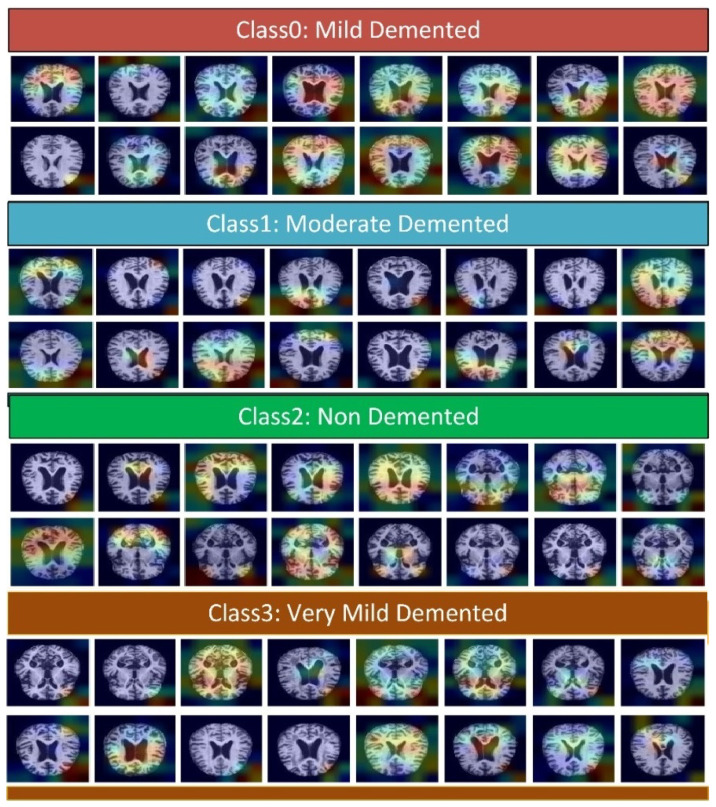
Generalization of the class activation map to locate the discriminative region through Grad-CAM.

**Table 1 molecules-27-07085-t001:** A literature evaluation of current cutting-edge approaches used in AD detection and classification.

Approach	Method	Accuracy	AUC	Precision	Recall	F1-Score	Imbalance Handling
Shereen [29]	DenseNet201 + ResNet101	94.86%	-	89.47%	83.74%	95.36%	None
Badiea [38]	AlexNet + ResNet50	94.8%	99.7%	-	-	-	None
Pradhan [31]	DenseNet169 + VGG19	82.6%	86.7%	-	-	-	None
Vasukidevi [32]	CapsNet	94.3%	-	94.92%	95.89%	95.19%	None
Battineni [33]	CNN Model	83.3%	-	-	-	-	None
Suganthe [16]	Inception-ResNet-v2	79.12%	81.9%	70.64%	28.22%	39.91%	None
Jyoti [34]	InceptionV4 + ResNet+ ADNet	93.18%	-	94%	93%	92%%	Data Augmentation
Jyoti [34]	DenseNet121 +DenseNet161 +DenseNet169	93.18%	-	94%	93%	92%	None
Jyoti [35]	InceptionV4	73.75%	-	-	-	-	None

**Table 2 molecules-27-07085-t002:** The model summary and list of parameters used in the proposed CNN model.

Model Summary		
**Layer Type**	**Output Shape**	**Parameters**
Input Layer	(None, 150, 150, 3)	0
Conv Block 1	(None, 75, 75, 8)	224
Conv Block 2	(None, 37, 37, 16)	1168
Conv Block 3	(None, 18, 18, 32)	4640
Conv Block 4	(None, 9,9,64)	18,496
Conv Block 5	(None, 4, 4, 128)	73,856
Dropout	(None, 4, 4, 128)	0
Flatten	(None, 2048)	0
Dense Block 1	(None, 512)	1,049,088
Dense Layer	(None, 4)	2052
SOFTMAX Layer	(None, 4)	0
Total Parameters		1,149,524
Trainable Parameters		1,149,524
Non-Trainable Parameters		0

**Table 3 molecules-27-07085-t003:** Performance comparison of proposed CNN Model and other state-of-the-art algorithms.

Architecture	Dataset	Accuracy	AUC	Precision	Recall	F1-Score
Proposed DAD-Net (with ADASYN)	Kaggle	99.22%	99.90%	99.30%	99.14%	99.19%
Proposed DAD-Net (without ADASYN)	Kaggle	90.00%	98.41%	91.34%	87.34%	88.09%
DEMNET	Kaggle	95.23%	97%	96%	95%	95%
Conv-BLSTM (SMOTE)	ADNI	82%	91%	78%	88%	82%
Conv-BLSTM (GAIN)	ADNI	82%	90%	79%	82%	82%
VGG16	ADNI	95.73%	-	96.33%	96%	95%
AlexNet	Kaggle	92.20%	99.45%	-	94.50%	-
ResNet-50	Kaggle	93.10%	98.82%	-	92.25%	-
Inception ResNet v2	Kaggle	79.12%	81.90%	70.64%	28.22%	39.91%

## Data Availability

Not applicable.

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
