# Peer review of "DAD-Net: Classification of Alzheimer’s Disease Using ADASYN Oversampling Technique and Optimized Neural Network"

_molecules, 2022, doi:10.3390/molecules27207085_

Round 1

Reviewer 1 Report

As a natural application the authors used the deep neural network to study the Alzheimer’s Disease (AD), which is a a neurological brain disorder that can be detected with MRI. In this manuscript the classification task on the the early stages of Alzheimer’s disease is explored with the Kaggle MRI image dataset. Different from the past works in order to balance the dataset the authors used a resampleing technique to generate train and test sets. In this study the authors obtained perfect results, such as precision, recall, F1-score, Area Under the Curve (AUC). All of them are above 99%. The neural network, which is called DAD-Net is also compared against DEMENET and CNN Model. As a byproduct the authors also used Grad-CAM to extract the important part with large classification weight. The manuscript is written well. However I have two questions and a suggestion.

First generating synthetic data is people's usual operation during using the the AI to tackle the medical problems. Because there are too many shortcomings for the datasets. However in order to avoiding the data leakage people should avoid to treat all the synthetic samples as independent ones. If one put the synthetic samples from the same patient into both train and test set, a perfect result will be obtained undoubtly. I think the authors should present their resampling and dataset setting processes in detail to clarify no such danger of data leakage.

Second the DAD-Net's structure appears too much like the usual CNN to me. It surprises me that such perfect results can be obtained for the early stage AD. If the data operation is clear with the data leakage, I suggest the authors could spend more efforts on exploring the reason why the network is so powerful by comparing deep CNN.

Third as the author claimed the balancing dataset is crucial in this problem, I suggest the authors could do a experiment by using a dataset with straightforward equal weight resampling. If the classification results are much more worse than the results in the manuscript, the conclusion would be more convincible.

After convincing answers to the questions in above I think the manuscript will be suitable to publish.

Author Response

Thank you very much for your insightful and valuable comments, we have attached the "Author's Response to Reviewers-1".

Reviewer 2 Report

(1) ADNI is a widely used dataset in AD research, it is recommended to test algorithms on this dataset (the author maybe need to transform the 3D image to 2D image)

(2) If the model trained on Kaggle has the same performance on ADNI or another dataset?

(3) In Figure 14, if the proposed method has good performance than DEMNET?  why?  Accuracy?

(4) What is the meaning of multiple curves in figure 15?  

(5) Figure 16 and Table 3, is the accuracy of DEMENT and DAD-NET consistent?

(6) The methods listed in table 3 should be compared with the proposed method.

(7) If the ADASYN is proposed by the author? The authors should compare the proposed net with other methods rather than compare the methods with or without ADASYN. The comparisons among different methods without ADASYN are unnecessary.    

Author Response

Thank you very much for your insightful and valuable comments, we have attached the "Author's Response to Reviewers-2".

Reviewer 3 Report

In this work, the authors present a CNN classifier for Alzheimer’s disease by using ADASYN oversampling. Furthermore, it shows the better performance than the previous models. Also, this paper is clearly written in a good language, thoroughly prepared. However, this manuscript bears some intrinsic disadvantages, thus the paper cannot be recommended. Please find the reasons below.

1.     The most innovation in this study is ADASYN oversampling, which is able to solve the problem for imbalanced dataset. Nevertheless, ADASYN oversampling has been adopted for a couple of times for CNN based classification of Alzheimer’s disease in the previous studies (Feng and Li 2021; Koh et al. 2020), thus the presented classier is not innovative enough.

2.     The authors implemented Grad-CAM algorithms to visualize the classification. However, the authors did not explain what the focus of CNN is and why the heat maps are like these.

3.     In table 1, the authors presented some comparisons for different models. However, are they based on the same dataset? If not, is that reasonable to directly compare them?

4.     The authors are not careful enough. For example, they leveraged repeated sentences in the manuscript. Specifically, in row 215, they presented “the primary idea behind ADASYN is to use a weighted distribution for different minority class…”; in row 219, they presented a same sentence.

Author Response

(The authors gave the same response as above.)

Round 2

Reviewer 3 Report

1.     The author claim that the innovation of the manuscript is the high accuracy. However, Oversampling’s most important drawback is that it makes exact copies of existing examples, which could lead to the model generate better that it actually does. In other words, if the features of small dataset can be well captured by the model, the accuracy can be definitely improved after oversampling (which is essentially overfitting). Therefore, I don’t think it is as good as one that was shown in the paper (with oversampling, the accuracy only improves by 3.49% in the present model). Furthermore, if the authors want to validate their results, they should conduct cross-validation, which is not adopted in the manuscript. Before that, it cannot convince me.

2.     The authors seem not very familiar with Grad-CAM algorithms. By using this algorithm, our aim is to make the classification more reasoning. For example, in this webpage “https://towardsdatascience.com/understand-your-algorithm-with-grad-cam-d3b62fce353”, the authors leverage Grad-CAM algorithm to recognize an image of cat, wherein cat’s whiskers, ears, and eyes are emphasized in shades of red and yellow while the rest of the image is colored in blues and greens. In other words, it is these features that distinguish cats with other categories. In the same ways, the authors should analyze what are the key features in the images to classify the disease.

3.     The authors claimed that the models come from same database. However, it is not the truth. For example, Suganthe et al. and Raju et al’s models are based on Kaggle database [1,2] while Islam and Zhang’s model is based on OASIS database [3].

4.     The authors are still not careful enough. For example, the reference 52 is Dozat’s study, whereas authors indicate that it is Vasukidevi’s study.

Reference

[1]      R. C. Suganthe, M. Geetha, G. R. Sreekanth, K. Gowtham, S. Deepakkumar, R. Elango, NVEO-NATURAL VOLATILES Essent. OILS Journal| NVEO 2021, 8, 145.

[2]      M. Raju, M. Thirupalani, S. Vidhyabharathi, S. Thilagavathi, IOP Conf. Ser. Mater. Sci. Eng. 2021, 1084, 012017.

[3]      J. Islam, Y. Zhang, Brain Informatics 2018, 5, DOI 10.1186/s40708-018-0080-3.

Author Response

Thank you for your insightful and valuable comments on the manuscript, a point-by-point response to the comments is attached.
